# Differences in Neuropathology between Nitroglycerin-Induced Mouse Models of Episodic and Chronic Migraine

**DOI:** 10.3390/ijms25073706

**Published:** 2024-03-26

**Authors:** Songyi Park, Harry Jung, Sang-Won Han, Sang-Hwa Lee, Jong-Hee Sohn

**Affiliations:** 1Institute of New Frontier Research Team, College of Medicine, Hallym University, Chuncheon 24252, Republic of Korea; songyip3697@gmail.com (S.P.); harry_88@naver.com (H.J.); rabiting@hallym.or.kr (S.-W.H.); bleulsh@naver.com (S.-H.L.); 2Department of Neurology, Chuncheon Sacred Heart Hospital, Hallym University College of Medicine, Chuncheon 24252, Republic of Korea

**Keywords:** nitroglycerin, episodic migraine, chronic migraine, trigeminal spinal subnucleus caudalis, anterior cingulate cortex, vasoactive intestinal peptide, calcitonin gene-related peptide, pituitary adenylate cyclase-activating peptide, substance P

## Abstract

Multiple animal models of migraine have been used to develop new therapies. Understanding the transition from episodic (EM) to chronic migraine (CM) is crucial. We established models mimicking EM and CM pain and assessed neuropathological differences. EM and CM models were induced with single NTG or multiple injections over 9 days. Mechanical hypersensitivity was assessed. Immunofluorescence utilized c-Fos, NeuN, and Iba1. Proinflammatory and anti-inflammatory markers were analyzed. Neuropeptides (CGRP, VIP, PACAP, and substance P) were assessed. Mechanical thresholds were similar. Notable neuropathological distinctions were observed in Sp5C and ACC. ACC showed increased c-Fos and NeuN expression in CM (*p* < 0.001) and unchanged in EM. Sp5C had higher c-Fos and NeuN expression in EM (*p* < 0.001). Iba1 was upregulated in Sp5C of EM and ACC of CM (*p* < 0.001). Proinflammatory markers were strongly expressed in Sp5C of EM and ACC of CM. CGRP expression was elevated in both regions and was higher in CM. VIP exhibited higher levels in the Sp5C of EM and ACC of CM, whereas PACAP and substance P were expressed in the Sp5C in both models. Despite similar thresholds, distinctive neuropathological differences in Sp5C and ACC between EM and CM models suggest a role in the EM to CM transformation.

## 1. Introduction

Migraine is a complex neurovascular disorder characterized by recurrent headache attacks and other associated symptoms. Animal models of migraine have been widely used to investigate mechanisms underlying the pathophysiology of migraine and obtain insights to guide the development of specific therapeutics. Several animal models of migraine have been described, some of which have proven valuable in developing novel therapeutic targets.

Episodic migraine (EM) and chronic migraine (CM) are both within the spectrum of migraine disorders but represent distinct clinical entities [1]. The relationship between EM and CM is complex. EM progresses to CM at an annual rate of 2.5%, but CM can also revert to EM [2,3]. The process, often referred to as migraine transformation or chronification, clinically manifests as a sustained increase in migraine frequency. It often leads to a persistent migraine state characterized by frequent and debilitating headaches, along with related symptoms. This escalation in migraine frequency results in high levels of disability, poor responses to treatment, and frequent recurrences in affected individuals. The corresponding decline in quality of life requires the provision of high-quality medical care and treatment [4].

A CM is classified as a single entity, which has led to the development of specific animal models mimicking its features for assessments of preventive medications and investigations of the pathophysiological mechanisms underlying migraine transformation [5,6,7,8,9]. Despite an incomplete understanding of the pathophysiological processes that lead from EM to CM, it remains critical to establish a clear distinction between these two conditions. However, considering that the underlying pathophysiology is likely to be similar and migraine frequency is likely to be influenced by complex polygenic factors, it remains unclear whether a clear distinction between EM and CM is useful [8].

Several mouse models of migraine have been developed, including transgenic mice and in vivo models of migraine-related pain through mechanical, electrical, or chemical stimulation. Each model has unique strengths and weaknesses [9]. In the study of migraine pathophysiology across multiple animal models, various experimental techniques have been used to observe neuronal activation in the spinal trigeminal nucleus caudalis (Sp5C). This is thought to involve the activation of trigeminal afferents, which densely innervate dural structures and project to second-order neurons in the trigeminal nucleus caudalis and the C1–C2 region of the spinal cord (trigeminocervical complex) [10]. As an EM progresses toward chronification into CM, we speculate that neuropathological changes occur in higher-level pain-modulation regions, upstream structures in the trigeminal pain pathway of migraine. The anterior cingulate cortex (ACC) is a key structure involved in various higher brain functions, including nociception, chronic pain, and emotions [11]. Clinical studies that have used neuroimaging and electrophysiologic exams have also found changes in the ACC in CM patients [12,13,14]. Therefore, to understand the pathogenesis of chronification from an EM to a CM, it is necessary to identify changes in two brain regions, the sp5C and ACC.

From a clinical perspective, there are obvious differences in clinical course and treatment response between an EM and a CM. Therefore, it is reasonable to establish animal models representing both an EM and a CM simultaneously and then analyze the differences in neuropathology associated with the pathological mechanisms unique to each condition. This approach provides a clear understanding of the differences between these conditions and the underlying pathological mechanisms responsible for the transformation from an EM to a CM. Here, we developed mouse models to replicate pain associated with an EM and a CM, then examined variations in neuropathology between these two models.

## 2. Results

### 2.1. Acute Hyperalgesia Was Highly Triggered by NTG Regardless of EM vs. CM

In contrast to the existing NTG-induced migraine model, we distinguished between episodic and chronic forms of migraine with EM and CM models differing in the number of NTG injections and disease-induction period. Specifically, NTG at a dose of 10 mg/kg was injected once to establish the EM model and five times every other day for 9 days to establish the CM model (Figure 1A). Evaluations of the mechanical sensitivities of the two models were performed via a behavioral test using von Frey filaments, and acute hyperalgesia was measured in the hind paws of the mice at 2 h after each NTG injection (Figure 1B). Intriguingly, the mechanical thresholds in the EM and CM models were significantly decreased compared with the VEH group (Figure 1C,D, and Table 1). These results indicated that acute hyperalgesia is triggered by NTG injection in both EM and CM models, regardless of the number or duration of NTG injections.

### 2.2. Neural Activation in the EM and CM Models Showed Substantial Differences in the Sp5C and ACC

A behavioral evaluation comparing acute hyperalgesia did not show any significant pathophysiological differences between EM and CM models (Figure 1 and Table 1). However, we aimed to clearly distinguish each condition’s unique pathological mechanisms and corresponding neuropathological differences. The degree of neural activation was evaluated by examining c-Fos and NeuN expression patterns in the Sp5C and ACC regions of the EM and CM models using fluorescence microscopy (Figure 2). There were substantial differences in the distributions of c-Fos+ cells and c-Fos+NeuN+ cells in each brain region between the EM and CM models. In the Sp5C, neural activation was increased in the NTG-induced migraine groups compared with the VEH group, and the numbers of c-Fos+ cells and c-Fos+NeuN+ cells were significantly greater in the EM model than in the CM model (Figure 2A–C and Table 1). Surprisingly, in the ACC region, the numbers of c-Fos+ cells and c-Fos+NeuN+ cells were significantly increased in the CM group compared with the VEH and EM groups (Figure 2D–F and Table 1). Therefore, rather than numerically comparing the degree of acute hyperalgesia in EM and CM models, we sought to clearly identify the pathological mechanism based on neural activation in each brain region.

### 2.3. Microgliosis in Sp5C and ACC Brain Regions Differed between EM and CM Models

Our results showed differences in neural activation between EM and CM models, and a previous study demonstrated that neural activation was closely associated with microgliosis [15]. Iba1 expression was examined using fluorescence microscopy to evaluate microgliosis in each brain region of both NTG-induced migraine models. In the Sp5C, Iba1+ cells were increased in the NTG-induced migraine groups compared with the VEH group, and there were significant differences between the EM and CM groups (Figure 3A,B, and Table 1). In the ACC, Iba1+ cells were significantly increased only in the CM group, compared with the VEH and EM groups (Figure 3C,D, and Table 1). The extent of microglial activation in each brain region differed between the EM and CM groups and correlated with neural activation.

**Figure 3 ijms-25-03706-f003:**
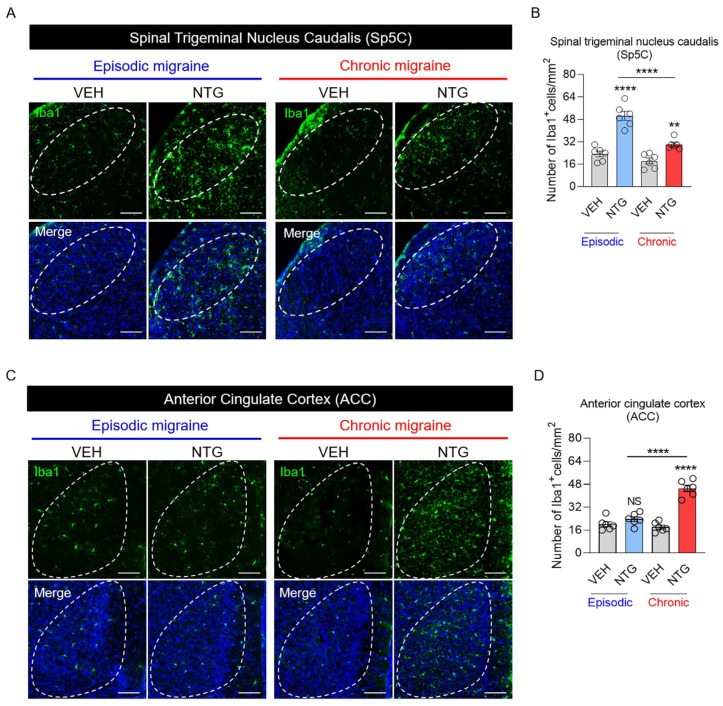
Iba1 expression patterns in mouse models of episodic and chronic NTG-induced migraine (EM and CM, respectively). (**A**) Representative images of Iba1 (green) immunofluorescence staining in the Sp5C of EM and CM mouse models. Scale bars = 20 μm. (**B**). Numbers of Iba1+ cells/mm^2^ in the Sp5C of EM and CM mouse models. (**C**) Representative images of Iba1 (green) immunofluorescence staining in the ACC of EM and CM mouse models. Scale bars = 20 μm. In the merged results, DAPI was used along with the Iba1 marker (**A**,**C**). (**D**) Numbers of Iba1+ cells/mm^2^ in the ACC of EM and CM mouse models. One-way ANOVA with post hoc Tukey test: ** *p* < 0.01; **** *p* < 0.0001; ns, not significant. Abbreviations: ACC, anterior cingulate cortex; NTG, nitroglycerin; Sp5C, spinal trigeminal nucleus caudalis; and VEH, vehicle control.

**Table 1 ijms-25-03706-t001:** Results of mechanical hypersensitivity measurement using the von Frey filament test and statistical analysis of neurons and microglial cell counts in mouse models (EM and CM, respectively), *** *p* < 0.001; and **** *p* < 0.0001.

Figure		Type of Analysis	Group	*N*	Mean ± SEM	F Value	*p* Value
Figure 1	1C	Non-parametric analysis (Mann–Whitney test)	VEH	9	1.044 ± 0.152	-	*** *p* = 0.0005
			NTG	8	0.293 ± 0.076		
	1D	Parametric analysis (Unpaired *t*-test)	VEH	9	1.311 ± 0.442	-	*** *p* = 0.0004
			NTG	10	0.189 ± 0.086		
Figure 2	2B	Parametric analysis (One-way ANOVA with Tukey’s multiple comparisons test)	EM_VEH	5	91.6 ± 3.28	22.94	**** *p* < 0.0001
			EM_NTG	5	147.8 ± 6.272		
			CM_VEH	5	93 ± 3.271		
			CM_NTG	5	116.6 ± 7.75		
	2C	Parametric analysis (One-way ANOVA with Tukey’s multiple comparisons test)	EM_VEH	5	60 ± 3.178	25.23	**** *p* < 0.0001
			EM_NTG	5	99.8 ± 6.061		
			CM_VEH	5	58.6 ± 2.462		
			CM_NTG	5	74 ± 2.236		
	2E	Parametric analysis (One-way ANOVA with Tukey’s multiple comparisons test)	EM_VEH	5	298.2 ± 8.587	172	**** *p* < 0.0001
			EM_NTG	5	323 ± 9.884		
			CM_VEH	5	296.2 ± 8.851		
			CM_NTG	5	577.8 ± 13.562		
	2F	Parametric analysis (One-way ANOVA with Tukey’s multiple comparisons test)	EM_VEH	5	112 ± 3.715	175.6	**** *p* < 0.0001
			EM_NTG	5	147.4 ± 3.043		
			CM_VEH	5	119.6 ± 5.519		
			CM_NTG	5	288.4 ± 10.068		
Figure 3	3B	Parametric analysis (One-way ANOVA with Tukey’s multiple comparisons test)	EM_VEH	6	23.33 ± 2.028	37.95	**** *p* < 0.0001
			EM_NTG	6	50.67 ± 3.273		
			CM_VEH	6	18.33 ± 2.092		
			CM_NTG	6	30.17 ± 1.447		
	3D	Parametric analysis (One-way ANOVA with Tukey’s multiple comparisons test)	EM_VEH	6	20.17 ± 1.740	48.03	**** *p* < 0.0001
			EM_NTG	6	23.67 ± 1.706		
			CM_VEH	6	18.17 ± 1.352		
			CM_NTG	6	45.17 ± 2.272		

### 2.4. Neuroinflammation in EM and CM Models Was Closely Associated with Neuronal Activation and Microgliosis in the Sp5C and ACC

A study showed that NTG-induced neuroinflammation causes migrainous hyperalgesia in specific brain regions, such as the Sp5C [16]. However, because our results showed differences in neural activation and microgliosis patterns in Sp5C and ACC regions between the EM and CM groups, neuroinflammation was analyzed under the same conditions and in the same regions using qRT-PCR and immunoblotting. First, inflammation-related activated markers (e.g., proinflammatory cytokines IL-1β, IL-6, TNF-α, and anti-inflammatory cytokines IL-4 and IL-10) were analyzed using qRT-PCR. The results indicated increased expression of proinflammatory cytokines (IL-1β, IL-6, and TNF-α) in the Sp5C regions of NTG-induced migraine groups compared with the VEH group; expression levels were significantly higher in the EM group than in the CM group. However, there were no significant differences among groups in the levels of anti-inflammatory cytokines (IL-4 and IL-10) (Figure 4A and Table 2). In the ACC, the expression levels of IL-1β, IL-6, and TNF-α were higher in the NTG-induced migraine groups than in the VEH group; proinflammatory cytokine expression was significantly higher in the CM group than in the EM group (Figure 4B and Table 2). Additionally, immunoblotting analysis of TNF-α, Iba1, and NF-κB p65 expression showed patterns similar to immunofluorescence staining and qRT-PCR, confirming the close relationship between neuroinflammation and microgliosis (Figure 4C).

### 2.5. Neuropeptide Expression Patterns in the Sp5C and ACC Differed between EM and CM Models

To investigate the dynamic differences in neuropeptide expression between NTG-induced models of EM and CM, we evaluated the expression levels of CGRP, VIP, PACAP, and substance P using immunofluorescence staining. The level of CGRP expression in the Sp5C was significantly higher in all NTG-induced migraine groups than in the VEH group, with higher expression in the CM model than in the EM model; however, there was no significant difference between the two NTG-induced migration groups (Figure 5A,B, and Table 3). In contrast, the level of VIP expression in the Sp5C was significantly elevated only in the EM model. PACAP levels were increased in the NTG-induced migraine groups compared with the VEH group, and the levels were significantly higher in the EM group than in the CM group. The level of substance P expression was higher in both NTG-induced EM and CM groups than in the VEH group, but no significant differences existed between the EM and CM groups. We also evaluated the expression levels of neuropeptides in the ACC. The level of CGRP expression significantly differed between the NTG-induced migraine groups and the VEH group, with higher expression in the CM model; there was no significant difference between the EM and CM models (Figure 6A,B, and Table 3). Additionally, VIP was expressed at higher levels in both EM and CM groups compared with the VEH group; the level was significantly higher in the CM group than in the EM group. Finally, the groups had no significant differences in PACAP or substance P expression levels in the ACC. In summary, four neuropeptides showed different levels of induction between brain regions and according to the stage of migraine (Figure 7).

## 3. Discussion

Although mechanical thresholds were similar in the EM and CM models in this study, there were notable neuropathological distinctions in the Sp5C and ACC between groups. Neuronal and microglial markers, including c-Fos, NeuN, and Iba1, showed significantly elevated expression in the Sp5C of the EM group and ACC of the CM group, compared with the VEH group. Proinflammatory cytokines, such as IL-1β, IL-6, and TNF-α, showed significantly increased expression in the Sp5C of the EM group and ACC of the CM group compared with the VEH group; there were no significant differences in the expression levels of anti-inflammatory cytokines, such as IL-4 and IL-10, between the groups. The expression levels of the neuropeptide CGRP were elevated in the Sp5C and the ACC in both EM and CM models, compared with controls. Levels of VIP expression were higher in the Sp5C of the EM group and the ACC of the CM group, whereas PACAP and substance P showed elevated expression levels in the Sp5C in both EM and CM groups compared with the controls (Figure 7).

Migraine is classified as either an EM or a CM. It is very rare for patients to present with a primary CM; up to 14% of patients with an EM are at risk of developing chronic daily headaches, particularly CM, within 1 year [17,18,19]. Patients with EM and CM exhibit differences in clinical presentation and response to traditional therapeutic treatments [20,21,22,23,24,25], and there may be physiological distinctions between EM and CM patients [26,27]. The overall understanding of migraine and the development of new treatments for its management have advanced through major translational research in humans and experimental animals. Despite the development and use of various preclinical models for migraine-related pain, further progress is required to improve patient quality of life. Additionally, despite clinical evidence regarding differences between patients with EM and patients with CM, there have been few studies to determine whether current animal models of migraine accurately represent EM and/or CM [28].

Various mouse models of migraine have been developed, including in vivo models induced by mechanical, electrical, or chemical stimuli and transgenic mice, each with unique advantages and limitations [9]. As one of the most common preclinical models for studying migraine-related pain, we used mice injected with NTG, which is converted to nitric oxide and vasoactive S-nitrosothiols. NTG is well tolerated, has a very short half-life, and can cross the blood–brain barrier with known and acceptable side effects [29]. We primarily focused on the Sp5C and ACC regions to examine differences in neuropathology between EM and CM models. We found distinct neuropathological differences between the two models in the Sp5C and ACC. Our results revealed increased neural activation and microgliosis in the Sp5C of the EM model and the ACC of the CM model. We also found that the degree of neuroinflammation was significantly greater in the Sp5C of the EM model and the ACC of the CM model.

The pain pathway in migraine involves input from the spinal trigeminal nucleus and rostral structures; the transmitted information is processed and integrated to generate a migraine headache. Peripheral sensory information is initially collected from the trigeminal nerve and relayed to the trigeminal ganglia, which consists of first-order neurons in the trigeminal system. Subsequently, the trigeminocervical complex, encompassing the Sp5C and the dorsal horn of the first cervical segments, functions as the second-order central nervous system relay within the trigeminal system and receives input from the trigeminal ganglia. The thalamus, functioning as the third-order relay within the trigeminal system, receives direct projections from the Sp5C; it modulates the activities of pain-related cortical regions, including the ACC, insular cortex, and primary and secondary somatosensory cortex [30]. The ACC, which receives sensory inputs from the thalamus and subcortical regions and projects sensory output to numerous regions (motor cortex, amygdala, midbrain regions, periaqueductal gray, rostral ventromedial medulla, and spinal dorsal horn), is a component of the endogenous opioid pain control circuit. It participates in the affective interpretation of pain, cognition, emotion, and motivation [31,32,33,34,35,36]. The neuropathological changes observed in the ACC of the NTG-induced mouse models were consistent with the evidence of ACC involvement from clinical studies involving patients with frequent headaches or CM. Structural analyses using high-resolution magnetic resonance imaging showed that ACC volume was significantly reduced in patients with migraine who exhibited more frequent headache attacks [37]. Functional neuroimaging studies in patients with CM have demonstrated increased functional connectivity between the ACC and other regions and changes in regional cerebral blood flow in the ACC [38,39]. As EM progressed toward chronification or transformed into CM because of recurrent headaches, we observed more pronounced neuropathological changes in the ACC, which is an upstream structure in the trigeminal pain pathway of migraine. Consequently, we speculate that the primary neuropathological changes occur in the Sp5C (the second-order relay of the trigeminal system) in the EM model and the ACC (a higher-level pain-modulation region within the trigeminal system) in the CM model.

Neuropeptides, including CGRP, VIP, PACAP, and substance P, have key roles in the pathophysiology of migraine. During trigeminovascular activation, CGRP, PACAP, VIP, and substance P are released; they act on vascular smooth muscle cells (to induce vasodilatation) and endothelial cells (to promote nitric oxide release) [40,41]. Substance P also contributes to plasma protein extravasation, while CGRP and PACAP are involved in peripheral and/or central sensitization, fundamental to migraine pathophysiology [42]. Furthermore, CGRP release triggers inflammatory mediator production, such as cytokines, beyond the nitric oxide-induced pathway [43]. Similar to our findings, previous studies indicated higher expression levels of proinflammatory cytokine genes or proteins (e.g., IL-1β, IL-6, and TNF-α) in the peripheral blood and trigeminovascular region of animals with NTG-induced migraine [43,44,45,46,47]. Given the pivotal role of CGRP in migraine pathophysiology, as evidenced by numerous clinical studies, monoclonal antibodies targeting these proteins have been developed for clinical practice. However, current therapies targeting CGRP, such as monoclonal antibodies that bind to CGRP or its receptor, are reportedly effective in only 50% to 60% of patients with migraine [48,49,50]. Therefore, the relationships of other neuropeptides (e.g., VIP, PACAP, and substance P) with the pathogenesis of migraine must be elucidated [42,51]. Among these neuropeptides, PACAP is the predominant isoform of PACAP-38 in nervous tissue, found in parasympathetic and sensory neurons of the trigeminal nucleus. PACAP modulates pain processing by increasing trigeminal nociceptor excitability through elevated levels of cyclic adenosine monophosphate [52]. Clinical studies have shown that PACAP levels during migraine attacks decrease following the administration of sumatriptan, a drug used to treat migraines. Additionally, PACAP injection has been reported to trigger headaches in migraine patients, prompting research into PACAP receptor inhibition as a potential migraine treatment strategy [53,54,55,56,57,58].

VIP, released from cranial parasympathetic preganglionic and cerebral perivascular nerves, acts as a potent vasodilator [51]. Although VIP infusion did not induce migraine attacks, VIP levels were elevated in patients with CM who exhibited increased cranial parasympathetic system activity during migraine attacks, as well as patients with EM and patients with CM during interictal periods. VIP may play a role in the chronification of migraine [59,60,61,62]. Parasympathetic activation can sensitize afferent nociceptors; this hypersensitivity and repeated stimulation may play a role in the conversion of EM to CM. VIP involvement in migraine chronification has been suggested [63]. Among the neuropeptides analyzed in the present study, only VIP exhibited significant differences in expression between EM and CM models, with higher expression in the Sp5C of the EM group and the ACC of the CM group. CGRP expression was significantly increased in both the Sp5C and ACC, whereas PACAP and substance P were strongly expressed in the Sp5C in both EM and CM models. However, data in the present study were generated in EM and CM animal models, with a focus on investigating differences in neuropathology between the two models. Therefore, it remains unclear whether differences in VIP expression are the cause or effect of migraine chronification; further research is required to elucidate the molecular mechanisms involved in migraine chronification or transformation. Moreover, further clinical studies are needed to gain a better understanding of the roles played by various neuropeptides in the pathogenesis of migraine.

To our knowledge, this is the first study to establish animal models representing both EM and CM simultaneously and then analyze differences in neuropathology associated with their underlying mechanisms. However, this study had some limitations. We primarily examined the results of behavioral tests and brain regions associated with pain processing. A migraine is characterized by recurrent attacks of pain and other associated symptoms. Previous preclinical and clinical studies focused on other components and related brain regions in patients with migraine and migraine animal models, including regions involved in emotional processing, cognitive components, and memories of pain processing. This study used only male mice to avoid the impact of hormonal variation on females [64], despite migraines being more prevalent in females. Further research should include both male and female mice and expand analyses to other brain regions and a broader range of behavioral tests to evaluate non-pain functions (e.g., cognition). Additionally, this study did not demonstrate pain reduction after the application of migraine drugs in animal models of EM and CM. Therefore, further research is needed to identify changes in EM and CM animal models after the use of various medications indicated for the treatment of migraine.

## 4. Materials and Methods

### 4.1. Animals

Specific pathogen-free male C57BL/6 mice, aged 7 weeks, were purchased from DooYeol Biotech (Seoul, Republic of Korea). The animals were housed in groups; they were acclimatized with access to food and water ad libitum under a 12 h light/dark cycle (light from 08:00 to 20:00) at a temperature of 23 °C ± 2 °C and relative humidity at 55% ± 10%. All experimental protocols received approval from the Institutional Animal Care and Use Committee (IACUC) of Hallym University (Hallym 2022-80, Chuncheon, Republic of Korea).

### 4.2. NTG-Induced Migraine Model

Nitroglycerin (NTG) prepared from a 0.974% solution in propylene glycol (Cat No. 1466506; USP, Rockville, MD, USA) was freshly diluted in 100 μL of saline, then intraperitoneally administered at a dose of 10 mg/kg during all in vivo experiments. The vehicle control (VEH) group received saline containing the same amount of propylene glycol (Cat No. 1576708; USP). The comparison between the saline-treated control and vehicle groups is presented in Appendix A and Table 4, showing no significant differences between the two groups.

### 4.3. Experimental Groups

Animals were divided into four groups:Episodic sham (EM-sham) group: mice received a single dose of saline with propylene glycol (*n* = 15);Episodic NTG-induced migraine (EM) group: mice received a single dose of NTG (10 mg/kg) (*n* = 15);Chronic sham (CM-sham) group: mice received five doses of saline with propylene glycol over 9 days (*n* = 15);Chronic NTG-induced migraine (CM) group: mice received five doses of NTG (10 mg/kg) over 9 days (*n* = 15).

The EM and CM models differed in the period and number of injections. In the EM model, animals were sacrificed 4 h after a single injection of NTG; in the CM model, animals received five injections of NTG over 9 days and were sacrificed 4 h after the last injection.

### 4.4. Behavioral Test for Mechanical Hyperalgesia

NTG-induced pain was measured at the threshold of 50% paw withdrawal responses to mechanical stimulation. In the EM model, von Frey filaments were used to evaluate the paw withdrawal response 2 h after NTG injection. In the CM model, the paw-withdrawal response was evaluated using von Frey filaments after each of the five NTG injections over a 9-day period. The mice were placed on a metal mesh floor under transparent acrylic cells without a floor and acclimatized for 15 min. To assess mechanical hyperalgesia, von Frey filaments (0.008, 0.02, 0.04, 0.07, 0.16, 0.4, 0.6, 1, 1.4, 2, 4, 6, 8,10, 26, 60, 100, 180, and 300 g) were pressed onto the plantar surface. The up-and-down paradigm began with the 0.6 g filament, and the threshold for a 50% paw withdrawal response was analyzed using the one-way analysis of variance (ANOVA).

### 4.5. Immunofluorescence Staining

Mice were anesthetized via intraperitoneal injection of 2.5% avertin, then perfused with 50 mL of phosphate-buffered saline or saline through the heart’s left ventricle. Whole brains were harvested, fixed with 4% paraformaldehyde for 16 h, dehydrated in 15% sucrose solution, and then placed in 30% sucrose solution until they sank to the bottom of the container. Mouse brain tissues were cut into 20–30 μm thick sections using a cryostat (Leica, Wetzlar, Germany). One of every five to seven slices was collected and stored at −80 °C. To examine neuron activation, brain tissue slices were permeabilized in 0.5% phosphate-buffered saline with Triton X-100 for 5 min, incubated in a blocking solution at room temperature (RT) for 1 h, and incubated with primary antibodies for double staining of c-Fos (Cat No. 226008, 1:200; Synaptic Systems, Göttingen, Germany) and NeuN (Cat No. ab104224, 1:200; Abcam, Cambridge, UK) at 4 °C for 16 h, then at RT for 1 h. To examine increases in microglial activation, staining of Iba1 (Cat No. ab22378, 1:200; Abcam) was performed at 4 °C for 16 h, then at RT for 1 h. To compare neurotransmitter changes in the trigeminal spinal nucleus caudalis (Sp5C) and anterior cingulate cortex (ACC) regions of the brain, the following markers were assessed via staining: vasoactive intestinal peptide (VIP) (Cat No. ab272726, 1:200; Abcam), calcitonin gene-related peptide (CGRP) (Cat No. ab81887, 1:200; Abcam), pituitary adenylate cyclase-activating polypeptide (PACAP) (Cat No. sc-166180, 1:200; Santa Cruz Biotechnology, Santa Cruz, CA, USA), and substance P (Cat No. ab14184, 1:200; Abcam) at 4 °C for 16 h, then at RT for 1 h. After the slides had been washed, they were incubated with the appropriate Alexa-Fluor 488- or 594-conjugated secondary antibody (1:500) at RT for 1 h. For nuclear staining, slides were incubated with 4′,6-diamidino-2-phenylindole (DAPI, 1:10,000) for 20 min before mounting. Immunofluorescence was visualized using fluorescence microscopy (Axio Scope 5 Laboratory microscope; Carl Zeiss, Oberkochen, Germany). Fluorescence intensity quantification and cell counting were performed using ImageJ software, 1.49v (NIH, Bethesda, MD, USA) or Photoshop version CS6 (Adobe Systems, San Jose, CA, USA). Representative data from two independent experiments are shown.

### 4.6. mRNA Expression

Brain tissue was harvested in a TRIzol reagent (Thermo Fisher Scientific, Waltham, MA, USA) and stored at −80 °C until processing. Total RNA was isolated from Sp5C and ACC tissues; cDNA was synthesized using M-MLV reverse transcriptase and oligo-dT primers (Invitrogen, Carlsbad, CA, USA). A reverse transcription polymerase chain reaction (RT-PCR) was performed with a cDNA template, primer, dNTP, 10× buffer, and Taq polymerase. Additionally, quantitative RT-PCR (qRT-PCR) was performed using SYBR Green premix (Enzynomics, Daejeon, Republic of Korea) in a real-time PCR detection system (Bio-Rad, Hercules, CA, USA). Representative data from two independent experiments are shown.

The primers used were as follows:

IL-1β, 5′-TTCACCATGGAATCCGTGTC-3′, 5′-GTCTTGGCCGAGGACTAAGG-3;

IL-6, 5′-CCTCTGGTCTTCTGGAGTACC-3′, 5′-ACTCCTTCTGTGACTCCAGC-3′;

TNF-α, 5′-TTCGAGTGACAAGCCTGTAG-3′, 5′CTTTGAGATCCATGCCGTTG-3;

IL-4, 5′-CAGCTAGTTGTCATCCTGCT-3′, 5′-ACCTCGTTCAAAATGCCGATG-3′;

IL-10, 5′-CTCTGATACCTCAGTTCCCA-3′, 5′-GTCCCCAATGGAAACAGCTT-3′;

GAPDH, 5′-CCTGTTGCTGTAGCCGTAT-3′, 5′-ACTCTTCCACCTTCGATGC-3′.

### 4.7. Immunoblotting

Protein lysates from Sp5C and ACC tissues (20 µg of protein) in mouse brains were separated by 5%–20% (*w*/*v*) gradient Bis-Tris sodium dodecyl sulfate–polyacrylamide gel electrophoresis (Cat No. 3450118; Bio-Rad) and transferred onto polyvinylidene difluoride membranes (Bio-Rad). After blocking in 5% bovine serum albumin for 1 h at RT, the membranes were incubated overnight at 4 °C with primary antibodies against TNF-α (Cat No. sc-52746; 1:1000; Santa Cruz Biotechnology), NF-κB (Cat No. 8242S, 1:1000; Cell Signaling Technology, Danvers, MA, USA), Iba1 (Cat No. 019-19741, 1:1000; Wako, Osaka, Japan), and β-actin (Cat No. 3700S; 1:1000; Cell Signaling Technology) in 5% bovine serum albumin. Blots were developed using appropriate horseradish peroxidase-linked secondary antibodies and observed using a chemiluminescent detection system (PerkinElmer, Waltham, MA, USA). Bands were quantified with a densitometer and normalized relative to β-actin.

### 4.8. Statistical Analysis

Data are presented as mean ± standard error of the mean (SEM). All statistical analyses were performed using GraphPad Prism software (version 8; GraphPad Software, La Jolla, CA, USA). For datasets suitable for parametric analysis, mean differences between groups were analyzed using a one-way ANOVA with post hoc assessment using the Tukey test or the unpaired Student’s *t*-test. For datasets requiring non-parametric analysis, the Mann–Whitney test and Kruskal–Wallis test with Dunn’s multiple comparisons test were conducted using Prism. Additionally, all results underwent reanalysis using either non-parametric or parametric analysis through SPSS (IBM SPSS Statistics 26). In all analyses, *p* < 0.05 was considered statistically significant.

## 5. Conclusions

This study demonstrated that mouse models representing both an EM and a CM, established simultaneously, exhibited notable neuropathological distinctions in the Sp5C and ACC after NTG injection. We observed increased neuronal activation and microgliosis in the Sp5C of the EM model and ACC of the CM model. Proinflammatory cytokines were also significantly upregulated in the Sp5C of the EM model and ACC of the CM model. Among the neuropeptides analyzed, only VIP exhibited higher levels in the Sp5C of the EM model and ACC of the CM model, highlighting significant differences between the two types of migraine. These changes may contribute to the underlying mechanisms driving an EM to a CM progression. Further preclinical studies are warranted, focusing on more advanced EM and CM animal models that allow migraine transformation or chronification investigations.

## Figures and Tables

**Figure 1 ijms-25-03706-f001:**
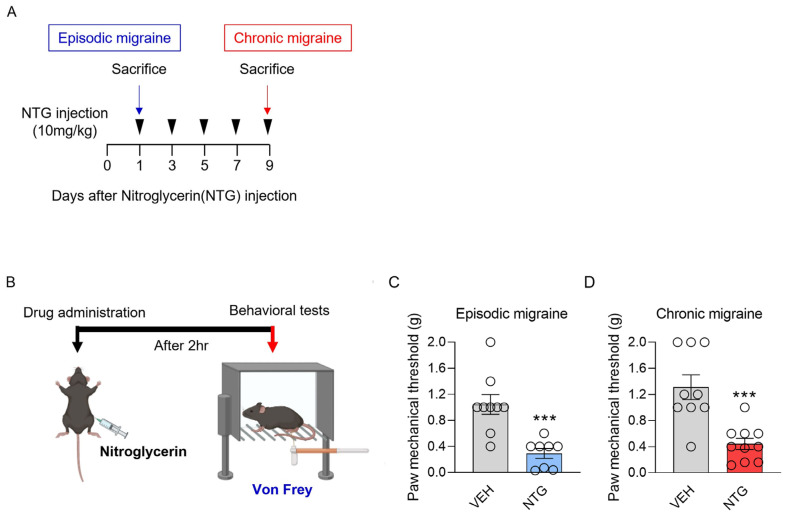
(**A**) Experimental protocols to establish mouse models of episodic and chronic NTG-induced migraine (EM and CM, respectively). (**B**–**D**) Experimental measurement of mechanical hypersensitivity using the von Frey filament test. (**C**) Non-parametric analysis (Mann–Whitney test) and (**D**) Parametric analysis (Unpaired *t*-test): *** *p* < 0.001. Abbreviations: NTG, nitroglycerin; VEH, vehicle control.

**Figure 2 ijms-25-03706-f002:**
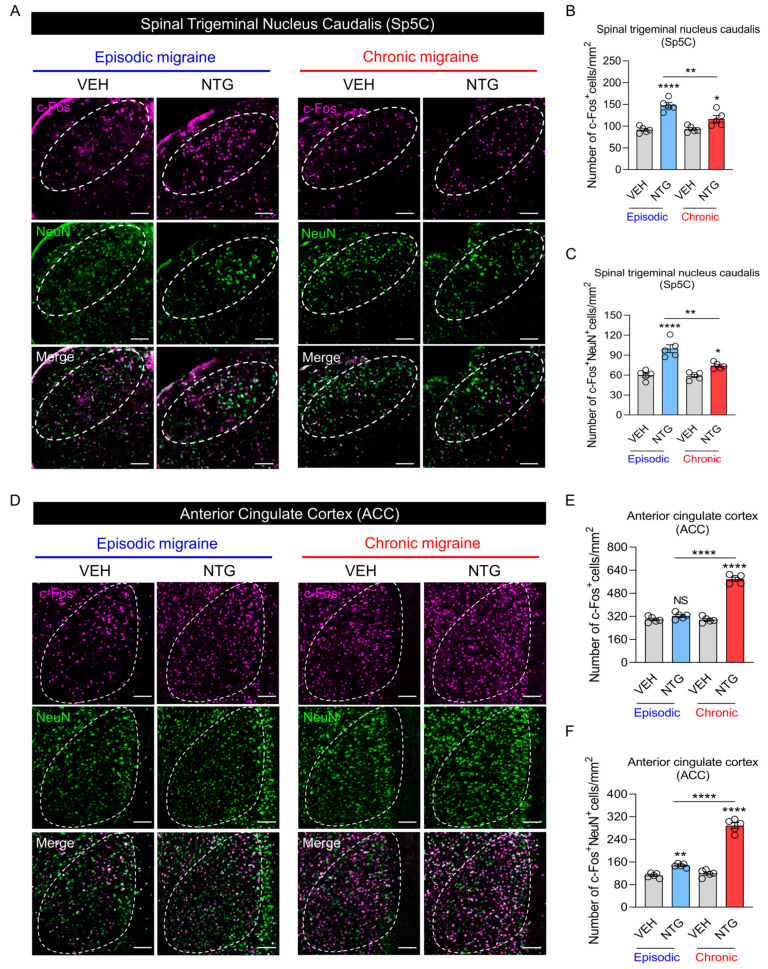
c-Fos and NeuN expression patterns in mouse models of episodic and chronic NTG-induced migraine (EM and CM, respectively). (**A**) Representative images of c-Fos (red) and NeuN (green) immunofluorescence staining in the Sp5C of EM and CM models. Scale bars = 10 μm. (**B**) Numbers of c-Fos+ cells/mm^2^ in the Sp5C of EM and CM mouse models. (**C**) Numbers of c-Fos+NeuN+ cells/mm^2^ in the Sp5C of EM and CM mouse models. (**D**) Representative images of c-Fos (red) and NeuN (green) immunofluorescence staining in the ACC of EM and CM mouse models. Scale bars = 10 μm. (**E**) Numbers of c-Fos+ cells/mm^2^ in the ACC of EM and CM mouse models. (**F**) Numbers of c-Fos+NeuN+ cells/mm^2^ in the ACC of EM and CM mouse models. One-way ANOVA with post hoc Tukey test: * *p* < 0.05; ** *p* < 0.01; **** *p* < 0.0001; ns, not significant. Abbreviations: ACC, anterior cingulate cortex; NTG, nitroglycerin; Sp5C, spinal trigeminal nucleus caudalis; and VEH, vehicle control.

**Figure 4 ijms-25-03706-f004:**
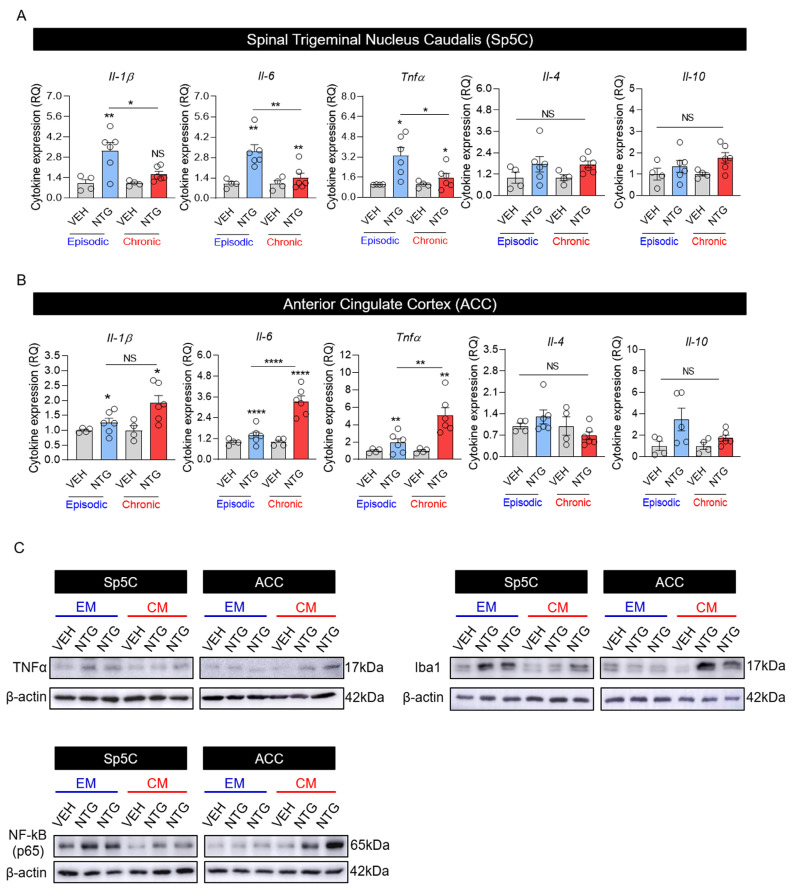
Inflammatory marker expression patterns in mouse models of episodic and chronic NTG-induced migraine (EM and CM, respectively). (**A**) Cytokine expression in the Sp5C of EM and CM mouse models, as determined using qRT-PCR. (**B**) Cytokine expression in the ACC of EM and CM mouse models, as determined using qRT-PCR. (**C**). TNF-α, Iba1, NF-κB, and β-actin (control) protein bands and relative expression, as determined using immunoblotting. Statistical analysis was conducted using the Kruskal–Wallis test with Dunn’s multiple comparisons test and one-way ANOVA with the post hoc Tukey test. * *p* < 0.05; ** *p* < 0.01; **** *p* < 0.0001; ns, not significant. Abbreviations: ACC, anterior cingulate cortex; NTG, nitroglycerin; Sp5C, spinal trigeminal nucleus caudalis; VEH, vehicle control.

**Figure 5 ijms-25-03706-f005:**
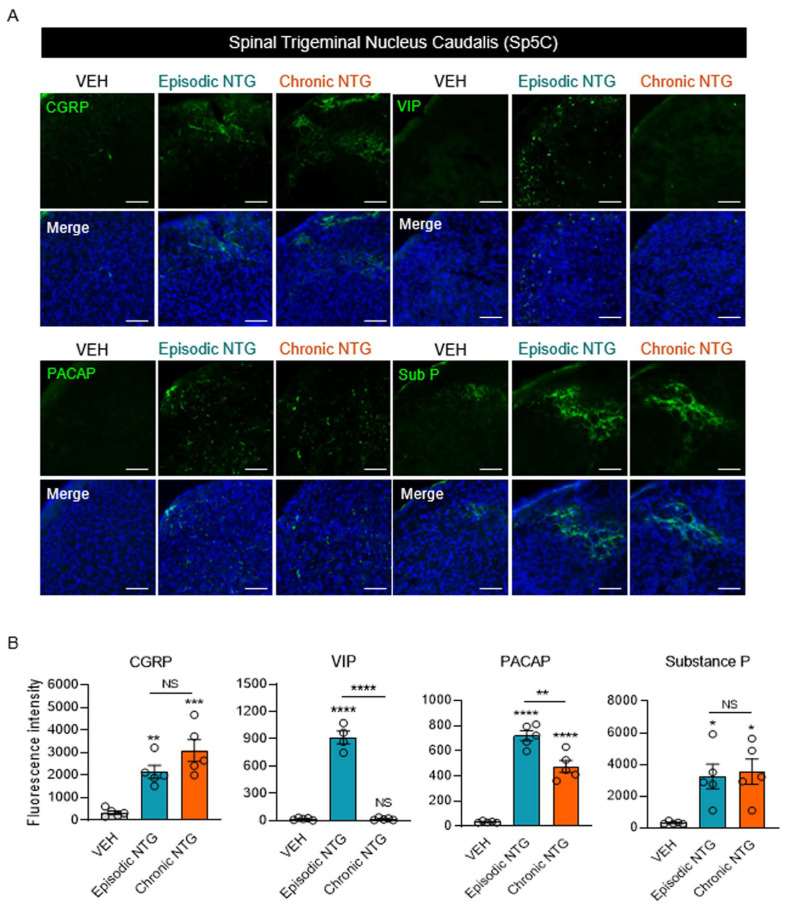
Neuropeptide expression patterns in the Sp5C in mouse models of episodic and chronic NTG-induced migraine (EM and CM, respectively). (**A**) Representative images of VIP, CGRP, PACAP, and substance P (green) immunofluorescence staining in the Sp5C of EM and CM mouse models. In the merged results, DAPI was added along with the VIP, CGRP, PACAP, and substance P markers. Scale bars = 20 μm. (**B**) Fluorescence intensities of VIP, CGRP, PACAP, and substance P in the Sp5C of EM and CM mouse models. One-way ANOVA with post hoc Tukey test: * *p* < 0.05; ** *p* < 0.01; *** *p* < 0.001; **** *p* < 0.0001. ns, not significant.

**Figure 6 ijms-25-03706-f006:**
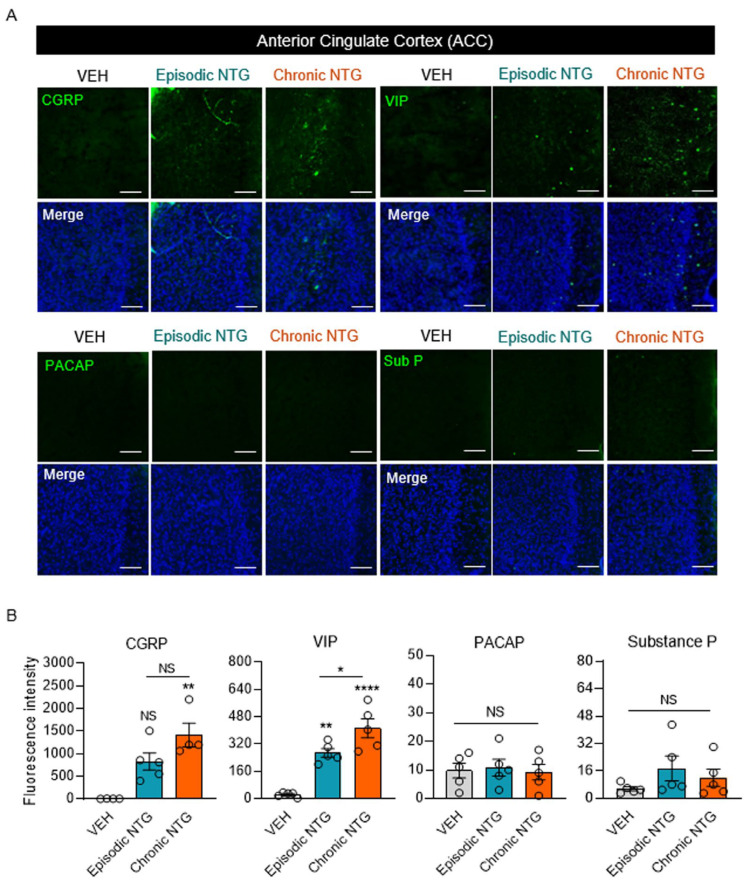
Neuropeptide expression patterns in the ACC in mouse models of episodic and chronic NTG-induced migraine (EM and CM, respectively). (**A**) Representative images of VIP, CGRP, PACAP, and substance P (green) immunofluorescence staining in the ACC of EM and CM mouse models. In the merged results, DAPI was added along with the VIP, CGRP, PACAP, and substance P markers. Scale bars = 20 μm. (**B**) Fluorescence intensities of VIP, CGRP, PACAP, and substance P in the ACC of EM and CM mouse models. Statistical analysis was conducted using the Kruskal–Wallis test with Dunn’s multiple comparisons test and one-way ANOVA with post hoc Tukey test: * *p* < 0.05; ** *p* < 0.01; **** *p* < 0.0001; ns, not significant. Abbreviations: ACC, anterior cingulate cortex; CGRP, calcitonin gene-related peptide; NTG, nitroglycerin; PACAP, pituitary adenylate cyclase-activating peptide; VEH, vehicle control; VIP, vasoactive intestinal peptide.

**Figure 7 ijms-25-03706-f007:**
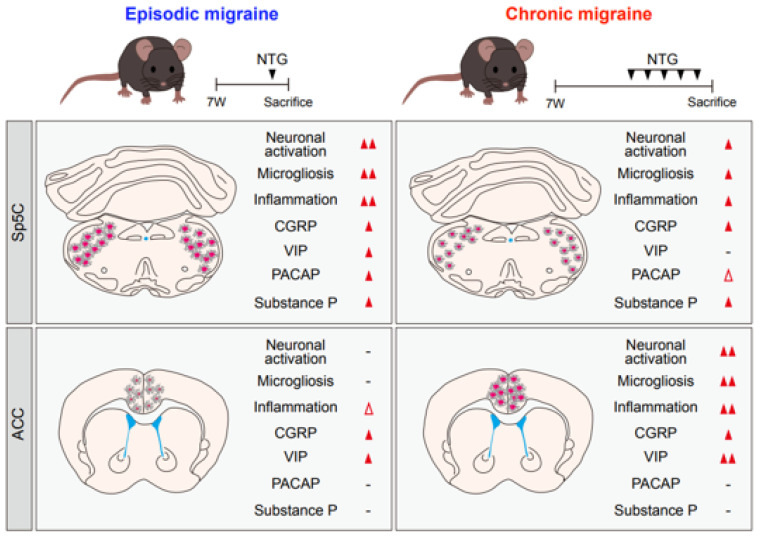
Summary of differences in neuropathology in the Sp5C and ACC between mouse models of episodic and chronic NTG-induced migraine (EM and CM, respectively). (-); ns, not significant, White triangle; slightly increase, Red triangle; significantly increase.

**Table 2 ijms-25-03706-t002:** Statistical analysis of inflammatory marker expression patterns in mouse models of episodic and chronic NTG-induced migraine (EM and CM, respectively), * *p* < 0.05; ** *p* < 0.01; *** *p* < 0.001; **** *p* < 0.0001.

Figure		Type of Analysis	Group	*N*	Mean ± SEM	F Value	*p*-Value
Figure 4A	IL-1β	Parametric analysis (One-way ANOVA with Tukey’s multiple comparisons test)	EM_VEH	4	1 ± 0.253	8.221	** *p* = 0.0015
			EM_NTG	6	3.25 ± 0.564		
			CM_VEH	4	1 ± 0.089		
			CM_NTG	6	1.63 ± 0.195		
	IL-6	Parametric analysis (One-way ANOVA with Tukey’s multiple comparisons test)	EM_VEH	4	1 ± 0.144	9.005	** *p* = 0.001
			EM_NTG	6	3.21 ± 0.468		
			CM_VEH	4	1 ± 0.207		
			CM_NTG	6	1.39 ± 0.333		
	Tnfα	Parametric analysis (One-way ANOVA with Tukey’s multiple comparisons test)	EM_VEH	4	1 ± 0.008	6.167	** *p* = 0.0055
			EM_NTG	6	3.3 ± 0.645		
			CM_VEH	4	1 ± 0.129		
			CM_NTG	6	1.54 ± 0.355		
	IL-4	Parametric analysis (One-way ANOVA with Tukey’s multiple comparisons test)	EM_VEH	4	1 ± 0.312	1.757	*p* = 0.1958
			EM_NTG	6	1.76 ± 0.428		
			CM_VEH	4	1 ± 0.172		
			CM_NTG	6	1.74 ± 0.195		
	IL-10	Parametric analysis (One-way ANOVA with Tukey’s multiple comparisons test)	EM_VEH	4	1 ± 0.276	2.126	*p* = 0.1371
			EM_NTG	6	1.37 ± 0.275		
			CM_VEH	4	1 ± 0.09		
			CM_NTG	6	1.76 ± 0.242		
Figure 4B	IL-1β	Parametric analysis (One-way ANOVA with Tukey’s multiple comparisons test)	EM_VEH	4	1 ± 0.038	5.467	** *p* = 0.0088
			EM_NTG	6	1.25 ± 0.151		
			CM_VEH	4	1 ± 0.159		
			CM_NTG	6	1.92 ± 0.25		
	IL-6	Parametric analysis (One-way ANOVA with Tukey’s multiple comparisons test)	EM_VEH	4	1 ± 0.086	23.87	**** *p* < 0.0001
			EM_NTG	6	1.39 ± 0.195		
			CM_VEH	4	1 ± 0.11		
			CM_NTG	6	3.33 ± 0.324		
	Tnfα	Parametric analysis (One-way ANOVA with Tukey’s multiple comparisons test)	EM_VEH	4	1 ± 0.119	10.72	*** *p* = 0.0004
			EM_NTG	6	1.92 ± 0.463		
			CM_VEH	4	1 ± 0.131		
			CM_NTG	6	5.09 ± 0.878		
	IL-4	Non-parametric analysis (Kruskal–Wallis test with Dunn’s multiple comparisons test)	EM_VEH	4	1 ± 0.094	-	*p* = 0.1372
			EM_NTG	6	1.3 ± 0.224		
			CM_VEH	4	1 ± 0.301		
			CM_NTG	6	0.69 ± 0.129		
	IL-10	Parametric analysis (One-way ANOVA with Tukey’s multiple comparisons test)	EM_VEH	4	1 ± 0.44	3.43	* *p* = 0.0444
			EM_NTG	6	3.47 ± 1.045		
			CM_VEH	4	1 ± 0.324		
			CM_NTG	6	1.75 ± 0.266		

**Table 3 ijms-25-03706-t003:** Statistical analysis of neuropeptide expression patterns in the Sp5C and ACC in mouse models of episodic and chronic NTG-induced migraine (EM and CM, respectively), * *p* < 0.05; *** *p* < 0.001; **** *p* < 0.0001.

Figure		Type of Analysis	Group	*N*	Mean ± SEM	F Value	*p* Value
Figure 5B	CGRP	Parametric analysis (One-way ANOVA with Tukey’s multiple comparisons test)	VEH	5	335.6 ± 79.75	18	*** *p* = 0.0002
			EM_NTG	5	2138.2 ± 286.3		
			CM_NTG	5	3087 ± 487.06		
	VIP	Parametric analysis (One-way ANOVA with Tukey’s multiple comparisons test)	VEH	5	18.6 ± 5.48	213.8	**** *p* < 0.0001
			EM_NTG	4	914 ± 69.38		
			CM_NTG	5	18.2 ± 5.49		
	PACAP	Parametric analysis (One-way ANOVA with Tukey’s multiple comparisons test)	VEH	5	34 ± 3.16	95.36	**** *p* < 0.0001
			EM_NTG	5	720 ± 39.55		
			CM_NTG	5	476.2 ± 47.23		
	Substance P	Parametric analysis (One-way ANOVA with Tukey’s multiple comparisons test)	VEH	5	325.2 ± 38.78	7.692	* *p* = 0.0071
			EM_NTG	5	3240.6 ± 778.48		
			CM_NTG	5	3558.8 ± 794.5		
Figure 6B	CGRP	Non-parametric analysis (Kruskal–Wallis test with Dunn’s multiple comparisons test)	VEH	4	7.75 ± 1.49	-	* *p* = 0.0009
			EM_NTG	5	828 ± 189.47		
			CM_NTG	4	1417 ± 263.57		
	VIP	Parametric analysis (One-way ANOVA with Tukey’s multiple comparisons test)	VEH	5	23.8 ± 6.11	30.49	**** *p* < 0.0001
			EM_NTG	5	268 ± 25.3		
			CM_NTG	5	411.8 ± 55.75		
	PACAP	Parametric analysis (One-way ANOVA with Tukey’s multiple comparisons test)	VEH	5	9.8 ± 2.58	0.08514	*p* = 0.9189
			EM_NTG	5	10.8 ± 2.96		
			CM_NTG	5	9.2 ± 2.76		
	Substance P	Parametric analysis (One-way ANOVA with Tukey’s multiple comparisons test)	VEH	5	5.6 ± 1.21	1.331	*p* = 0.3004
			EM_NTG	5	17.4 ± 7.24		
			CM_NTG	5	12 ± 4.97		

**Table 4 ijms-25-03706-t004:** Statistical analyses revealed no differences between the saline-treated control and vehicle groups.

Figure	Type of Analysis	Group	*N*	Mean ± SEM	F Value	*p*-Value
Appendix A	Parametric analysis (One-way ANOVA with Tukey’s multiple comparisons test)	CON	5	1.76 ± 0.75	2.54	*p* = 0.104
		EM_VEH	9	1.04 ± 0.15		
		CM_VEH	9	1.31 ± 0.19		

## Data Availability

The original contributions presented in the study are included in the article; further inquiries can be directed to the corresponding author.

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
