# Peer review of "Differences in Neuropathology between Nitroglycerin-Induced Mouse Models of Episodic and Chronic Migraine"

_ijms, 2024, doi:10.3390/ijms25073706_

Round 1
Reviewer 1 Report
Comments and Suggestions for Authors
The manuscript “Differences in Neuropathology Between Nitroglycerin-Induced Mouse Models of Episodic and Chronic Migraine“ by Songyi Park et al. is a research article trying to analyze the differences in neuropathology associated with the pathological mechanisms in episodic and chronic migraine. The experimental design provides an innovative approach by simultaneously studying acute and chronic migraine models, although most of its elements are known from previous research. The manuscript is well organized. The results were processed and presented in an appropriate manner. Methods and statistical analyses are appropriate. The references are adequate, but 24 of the 59 references are older than 10 years.
There are some suggestions to the authors:
Section 1 – Introduction
- It would be useful to add just a few sentences at the end of the introduction to explain why two specific regions (e Sp5C and ACC ) were evaluated and what information the selected markers provide. Some of the sentences from the Discusion can be used for this.
Section 2 – Results
- Page 2, lines 66-68: Behavioral test was performed 2 hours after each injection. Although the VEH group was used as a control, it would have been useful to test each animal before any treatment. Thus, the authors would have a baseline result for each animal and better and more accurate insight into the test results. According to Fig. 1C and 1D propylene glycol in VEH may also affect the results (depending on the dose). Therefore, it would have been useful if an additional saline-treated control group had been used.
- In the same lines (66-68) it was stated: “ Evaluations of the mechanical sensitivities of the two models were performed by a behavioral test using von Frey filaments, and acute hyperalgesia was measured in the hind paws of the mice at 2 h after each NTG injection (Fig. 1B).“ According to this, in Materials and Methods section (L 343-344) would be useful to change: “ In the CM model, the paw withdrawal response was evaluated using von Frey filaments after five NTG injections over a 9-day period.“ into “ In the CM model, the paw withdrawal response was evaluated using von Frey filaments after each of the five NTG injections over a 9-day period.“
- Page 3, line 103: Sentence starts with “A recent study…“ and the reference for this study is from 2012. I think that a more recent reference should be added or the word “recent“ should be removed from the sentence.
- Page 5, figure 2 – The c-Fos label is very poorly seen in red fluorescence images. The authors should try to change the shade of red color.
- Page 6, Figure 3 – There is no label on figures 3A and 3C saying what the blue fluorescence represents, and what was merged. It is DAPI but it is not indicated in the figure caption either. The blue signal is too strong (too long incubation?) and merging the two images does not contribute to the display of the results. Maybe it would be best to remove DAPI staining from the figure. The same remark applies to Figure 5A (page 8) and 6A (page 9).
- Page 3, lines 137-138: At the end of the sentence “In summary,….” add (Figure 7).
Section 4 – Materials and Methods
Animals (L 316) – According to the authors, only male animals were used. Since migraine affects females more than males, the suggestion to the authors is to include both male and female animals in their future research.
References – replace or add some newer references where possible.
Author Response
Mar 17, 2024
Reviewer 1
Int J Mol Sci
Dear Reviewer 1,
Please find attached a revised version of our manuscript, “Differences in Neuropathology Between Nitroglycerin-Induced Mouse Models of Episodic and Chronic Migraine” (ijms-2897306) for your review.
We appreciate your insightful feedback on the original submission. We have addressed most of your recommendations in this revised manuscript.
Details of the revisions made are outlined in sequence below, corresponding to your comments (presented in italics). We are pleased to resubmit this revised version for your consideration for publication in the International Journal of Molecular Sciences.
Comments to author:
The manuscript “Differences in Neuropathology Between Nitroglycerin-Induced Mouse Models of Episodic and Chronic Migraine“ by Songyi Park et al. is a research article trying to analyze the differences in neuropathology associated with the pathological mechanisms in episodic and chronic migraine. The experimental design provides an innovative approach by simultaneously studying acute and chronic migraine models, although most of its elements are known from previous research. The manuscript is well organized. The results were processed and presented in an appropriate manner. Methods and statistical analyses are appropriate. The references are adequate, but 24 of the 59 references are older than 10 years.
We thank you for your comments and suggestions, which have contributed to the improvement of our manuscript.
There are some suggestions to the authors:
Section 1 – Introduction
- It would be useful to add just a few sentences at the end of the introduction to explain why two specific regions (e Sp5C and ACC) were evaluated and what information the selected markers provide. Some of the sentences from the Discussion can be used for this.
Thank you for your comment. We have expanded the Introduction with additional sentences to provide further context and clarity, as follows:
Several mouse models of migraine have been developed, including transgenic mice and in vivo models of migraine-related pain through mechanical, electrical, or chemical stimulation. Each model has unique strengths and weaknesses [9]. In the study of migraine pathophysiology across multiple animal models, various experimental techniques have been used to observe neuronal activation in the spinal trigeminal nucleus caudalis (Sp5C). This is thought to involve the activation of trigeminal afferents, which densely innervate dural structures and project to second-order neurons in the trigeminal nucleus caudalis and the C1–C2 region of the spinal cord (trigeminocervical complex) [10]. As EM progresses toward chronification into CM, we speculate that neuropathological changes occur in higher-level pain modulation regions, upstream structures in the trigeminal pain pathway of migraine. The anterior cingulate cortex (ACC) is a key structure involved in various functions of the higher brain, including nociception, chronic pain, and emotions [11]. Clinical studies that have used neuroimaging and electrophysiologic exams have also found changes in the ACC in CM patients [12,13]. Therefore, to understand the pathogenesis of chronification from EM to CM, it is necessary to identify changes in two brain regions, the sp5C and ACC
(page 2, lines 52 – page 2, lines 65)
We have also updated the references as follows.
- Harriot, A.M., Strother, L.C., Vila-Pueyo, M., Holland, P.R.; Animal models of migraine and experimental techniques used to examine trigeminal sensory processing. JHP 2019,20,91, https://doi.org/10.1186/s10194-019-1043-7.
- Akerman, S., Holland, P.R., Hoffmann, J.; Pearls and pitfalls in experimental in vivo models of migraine: Dural trigeminovascular nociception. Cephalalgia 2013,33, 8, 577-592. https://doi.org/10.1177/0333102412472071.
- Tsuda, M., Koga, K., Chen, T., Zhuo, M.; Neuronal and microglial mechanisms for neuropathic pain in the spinal dorsal horn and anterior cingulate cortex. J Neurochem 2017, 141(4), 486–498. doi: 10.1111/jnc.14001.
- Jia, Z., Yu, S.; Grey matter alterations in migraine: a systematic review and meta-analysis. NeuroImage Clin 2017, 14, 130–140. doi: 10.1016/j.nicl.2017.01.019.
- de Tommaso, M., Losito, L., Difruscolo, O., Libro, G., Guido, M., Livrea, P.; Changes in cortical processing of pain in chronic migraine. Headache 2005, 45, 1208–1218. DOI: 10.1111/j.1526-4610.2005.00244.x
Section 2 – Results
- Page 2, lines 66-68: Behavioral test was performed 2 hours after each injection. Although the VEH group was used as a control, it would have been useful to test each animal before any treatment. Thus, the authors would have a baseline result for each animal and better and more accurate insight into the test results. According to Fig. 1C and 1D propylene glycol in VEH may also affect the results (depending on the dose). Therefore, it would have been useful if an additional saline-treated control group had been used.
Data comparing the saline-treated control group and the vehicle group are presented in Supplementary Figure 1 and Table 4, indicating no significant differences between the two groups. This information has been incorporated into the Materials and Methods section as follows:
The comparison between the saline-treated control group and the vehicle group is presented in Supplementary Figure 1 and Table 4, showing no significant differences between the two groups.
(page 16 , lines 362 – page 16, lines 364)
- In the same lines (66-68) it was stated: “ Evaluations of the mechanical sensitivities of the two models were performed by a behavioral test using von Frey filaments, and acute hyperalgesia was measured in the hind paws of the mice at 2 h after each NTG injection (Fig. 1B).“ According to this, in Materials and Methods section (L 343-344) would be useful to change: “ In the CM model, the paw withdrawal response was evaluated using von Frey filaments after five NTG injections over a 9-day period.“ into “ In the CM model, the paw withdrawal response was evaluated using von Frey filaments after each of the five NTG injections over a 9-day period.“
In response to your suggestion, we have modified the sentence to read: “In the CM model, the paw-withdrawal response was evaluated using von Frey filaments after each of the five NTG injections over a 9-day period.” This revision is now reflected in the Materials and Methods section as follows:
In the CM model, the paw-withdrawal response was evaluated using von Frey filaments after each of the five NTG injections over a 9-day period.
(page 17, lines 381 – page 17, lines 382).
- Page 3, line 103: Sentence starts with “A recent study…“ and the reference for this study is from 2012. I think that a more recent reference should be added or the word “recent“ should be removed from the sentence.
Thank you for your insightful comments. We have removed the term ‘recent’ from the revised manuscript as follows.
A study showed that NTG-induced neuroinflammation causes migrainous hyperalgesia in specific brain regions, such as the Sp5C.
(page 3, lines 119– page 3, lines 120).
- Page 5, figure 2 – The c-Fos label is very poorly seen in red fluorescence images. The authors should try to change the shade of red color.
To enhance clarity and visualization, we have changed the c-Fos staining results to violet color in Figure 2.
- Page 6, Figure 3 – There is no label on figures 3A and 3C saying what the blue fluorescence represents, and what was merged. It is DAPI but it is not indicated in the figure caption either. The blue signal is too strong (too long incubation?) and merging the two images does not contribute to the display of the results. Maybe it would be best to remove DAPI staining from the figure. The same remark applies to Figure 5A (page 8) and 6A (page 9).
We have combined all images with DAPI staining, and the brightness of DAPI has been uniformly adjusted across all figures. Furthermore, we have included the relevant information in the legends of Figures 3, 5, and 6 as follows.
In the merged results, DAPI was used along with the Iba1 marker (A, C).
(page 6, lines 181– page 6, lines 182)
In the merged results, DAPI was added along with the VIP, CGRP, PACAP, and substance P markers.
(page 10, lines 207– page 10, lines 208)
In the merged results, DAPI was added along with the VIP, CGRP, PACAP, and substance P markers.
(page 11, lines 217– page 11, lines 218)
- Page 3, lines 137-138: At the end of the sentence “In summary,….” add (Figure 7).
We have added “(Figure 7)” as follows
In summary, four neuropeptides showed different levels of induction between brain regions and according to the stage of migraine (Figure 7).
(page 3, lines 153– page 3, lines 155 )
Section 4 – Materials and Methods
Animals (L 316) – According to the authors, only male animals were used. Since migraine affects females more than males, the suggestion to the authors is to include both male and female animals in their future research.
We appreciate your feedback. Based on previous studies, we used male animal models in this research due to the potential variability introduced by hormonal changes in female mice. In our subsequent studies, we aim to compare the phenotypes between female and male mice. Accordingly, we have added a statement to the limitations section of the Discussion section as follows.
This study used only male mice to avoid hormonal variation impact in females [64], despite migraine being more prevalent in females. Further research should include both male and female mice and expand analyses to other brain regions and a broader range of behavioral tests to evaluate non-pain functions (e.g., cognition).
(page 16, lines 343– page 16, lines 346)
References – replace or add some newer references where possible.
Thank you for your suggestion. We have included additional references as follows:
- Harriot, A.M., Strother, L.C., Vila-Pueyo, M., Holland, P.R.; Animal models of migraine and experimental techniques used to examine trigeminal sensory processing. JHP 2019,20,91, https://doi.org/10.1186/s10194-019-1043-7.
- Akerman, S., Holland, P.R., Hoffmann, J.; Pearls and pitfalls in experimental in vivo models of migraine: Dural trigemi-novascular nociception. Cephalalgia 2013,33, 8, 577-592. https://doi.org/10.1177/0333102412472071.
- Tsuda, M., Koga, K., Chen, T., Zhuo, M.; Neuronal and microglial mechanisms for neuropathic pain in the spinal dorsal horn and anterior cingulate cortex. J Neurochem 2017, 141(4), 486–498. doi: 10.1111/jnc.14001.
- Jia, Z., Yu, S.; Grey matter alterations in migraine: a systematic review and meta-analysis. NeuroImage Clin 2017, 14, 130–140. doi: 10.1016/j.nicl.2017.01.019.
- de Tommaso, M., Losito, L., Difruscolo, O., Libro, G., Guido, M., Livrea, P.; Changes in cortical processing of pain in chronic migraine. Headache 2005, 45, 1208–1218. DOI: 10.1111/j.1526-4610.2005.00244.x
- Wu, S., Ren, X., Zhu, C. et al. A c-Fos activation map in nitroglycerin/levcromakalim-induced models of migraine. J Head-ache Pain 2022,23,128, https://doi.org/10.1186/s10194-022-01496-8.
We have addressed all the issues highlighted by the reviewers. We are grateful for the constructive feedback provided during the review process. We believe that our manuscript has been improved by these revisions.
Yours faithfully,
Jong-Hee Sohn, M.D. Ph.D.
Department of Neurology, Chuncheon Sacred Heart Hospital, Hallym University College of Medicine, 77 Sakju-ro, Chuncheon-si, Gangwon-state, 24253, Republic of Korea
Tel: +82-33-252-9970, Fax: +82-33-241-8063
E-mail: deepfoci@hallym.or.kr

Reviewer 2 Report
Comments and Suggestions for Authors
The fulfillment of the assumptions necessary for the application of the specified statistical test has not been indicated, and the validity of using the chosen post-hoc test has not been determined. Due to the low sample size, it is recommended to use a nonparametric equivalent of the applied statistical test.
The results of the statistical test are not reported according to the appropriate format, such as F(2;35) = 4.56; p = 0.02. Another example: H = 3.45; p = 0.02. The effect size for the applied statistical test has not been calculated. Relying solely on the p-value is insufficient.
Descriptive statistics for such a small sample size are inadequate. Additional statistics, such as the median, first and third quartiles, should be provided. The figures are entirely unreadable. The results of statistical tests, along with the effect size and additional descriptive statistics, should be included in a table to enhance manuscript clarity.
The proper approval number from the Ethics Committee for conducting this study is missing.
The introduction is written very superficially, with only a few references placed in square brackets. It does not provide specific data on what has been published so far and what is novel in this article.
Comments on the Quality of English LanguageModerate editing of English language required.
Author Response
Mar 17, 2024
Reviewer 2
Int J Mol Sci
Dear Reviewer 2,
Please find attached a revised version of our manuscript, “Differences in Neuropathology Between Nitroglycerin-Induced Mouse Models of Episodic and Chronic Migraine” (ijms-2897306) for your review.
We appreciate your insightful feedback on the original submission. We have addressed most of your recommedations in this revised manuscript.
Details of the revisions made outlined in sequence, corespodending to your comments (presented in italics). We are pleased to resubmit this revised version for your consideration for publication.
Comments to author:
The fulfillment of the assumptions necessary for the application of the specified statistical test has not been indicated, and the validity of using the chosen post-hoc test has not been determined. Due to the low sample size, it is recommended to use a nonparametric equivalent of the applied statistical test.
We thank you for your comments and suggestions, which have contributed to the improvement of our manuscript.
The results of the statistical test are not reported according to the appropriate format, such as F(2;35) = 4.56; p = 0.02. Another example: H = 3.45; p = 0.02. The effect size for the applied statistical test has not been calculated. Relying solely on the p-value is insufficient.
Descriptive statistics for such a small sample size are inadequate. Additional statistics, such as the median, first and third quartiles, should be provided. The figures are entirely unreadable. The results of statistical tests, along with the effect size and additional descriptive statistics, should be included in a table to enhance manuscript clarity.
We appreciate your valuable feedback. All results have been reanalyzed using either non-parametric or parametric analysis with SPSS (IBM SPSS Statistics 26). For non-parametric analysis, the Mann-Whitney test and Kruskal-Wallis test with Dunn’s multiple comparisons test were conducted using Prism. Parametric analysis was conducted using one-way ANOVA with Tukey’s multiple comparisons test. These results are shown in Tables 1–4. Additionally, the corresponding explanation has been included in the Materials and Methods section as follows:
For datasets suitable for parametric analysis, mean differences between groups were analyzed using one-way ANOVA with post hoc assessment using the Tukey test or the unpaired Student’s t-test. For datasets requiring non-parametric analysis, the Mann-Whitney test and Krus-kal-Wallis test with Dunn’s multiple comparisons test were conducted using Prism. Additionally, all results underwent reanalysis using either non-parametric or parametric analysis through SPSS (IBM SPSS Statistics 26). In all analyses, p < 0.05 was considered statistically significant.
(page 18, lines 443– page 18, lines 449)
The proper approval number from the Ethics Committee for conducting this study is missing.
The IACUC approval number has been included in the revised manuscript as follows:
All experimental protocols received approval from the Institutional Animal Care and Use Committee (IACUC) of Hallym University (Hallym 2022-80, Chuncheon, Republic of Korea).
(page 16, lines 355 – page 16, lines 357)
The introduction is written very superficially, with only a few references placed in square brackets. It does not provide specific data on what has been published so far and what is novel in this article.
Thank you for your comment. We have expanded the Introduction with additional sentences, as follows:
Several mouse models of migraine have been developed, including transgenic mice and in vivo models of migraine-related pain through mechanical, electrical, or chemical stimulation. Each model has unique strengths and weaknesses [9]. In the study of migraine pathophysiology across multiple animal models, various experimental techniques have been used to observe neuronal activation in the spinal trigeminal nucleus caudalis (Sp5C). This is thought to involve the activation of trigeminal afferents, which densely innervate dural structures and project to second-order neurons in the trigeminal nucleus caudalis and the C1–C2 region of the spinal cord (trigeminocervical complex) [10]. As EM progresses toward chronification into CM, we speculate that neuropathological changes occur in higher-level pain modulation regions, upstream structures in the trigeminal pain pathway of migraine. The anterior cingulate cortex (ACC) is a key structure involved in various functions of the higher brain, including nociception, chronic pain, and emotions [11]. Clinical studies that have used neuroimaging and electrophysiologic exams have also found changes in the ACC in CM patients [12,13]. Therefore, to understand the pathogenesis of chronification from EM to CM, it is necessary to identify changes in two brain regions, the sp5C and ACC
(page 2, lines 52 – page 2, lines 65)
Furthermore, we have added the references as follows:
- Harriot, A.M., Strother, L.C., Vila-Pueyo, M., Holland, P.R.; Animal models of migraine and experimental techniques used to examine trigeminal sensory processing. JHP 2019,20,91, https://doi.org/10.1186/s10194-019-1043-7.
- Akerman, S., Holland, P.R., Hoffmann, J.; Pearls and pitfalls in experimental in vivo models of migraine: Dural trigeminovascular nociception. Cephalalgia 2013,33, 8, 577-592. https://doi.org/10.1177/0333102412472071.
- Tsuda, M., Koga, K., Chen, T., Zhuo, M.; Neuronal and microglial mechanisms for neuropathic pain in the spinal dorsal horn and anterior cingulate cortex. J Neurochem 2017, 141(4), 486–498. doi: 10.1111/jnc.14001.
- Jia, Z., Yu, S.; Grey matter alterations in migraine: a systematic review and meta-analysis. NeuroImage Clin 2017, 14, 130–140. doi: 10.1016/j.nicl.2017.01.019.
- de Tommaso, M., Losito, L., Difruscolo, O., Libro, G., Guido, M., Livrea, P.; Changes in cortical processing of pain in chronic migraine. Headache 2005, 45, 1208–1218. DOI: 10.1111/j.1526-4610.2005.00244.x
We have addressed all the issues highlighted by the reviewers. We are grateful for the constructive feedback provided during the review process. We believe that our manuscript has been improved by these revisions.
Yours faithfully,
Jong-Hee Sohn, M.D. Ph.D.
Department of Neurology, Chuncheon Sacred Heart Hospital, Hallym University College of Medicine, 77 Sakju-ro, Chuncheon-si, Gangwon-state, 24253, Republic of Korea
Tel: +82-33-252-9970, Fax: +82-33-241-8063
E-mail: deepfoci@hallym.or.kr

Round 2
Reviewer 2 Report
Comments and Suggestions for Authors
The majority of recommendations have been implemented. In the future, I would suggest calculating appropriate effect size measures.
Comments on the Quality of English LanguageMinor editing of English language required